# A New Cryptic Lineage in Parmeliaceae (Ascomycota) with Pharmacological Properties

**DOI:** 10.3390/jof8080826

**Published:** 2022-08-08

**Authors:** Elisa Garrido-Huéscar, Elena González-Burgos, Paul M. Kirika, Joël Boustie, Solenn Ferron, M. Pilar Gómez-Serranillos, Helge Thorsten Lumbsch, Pradeep K. Divakar

**Affiliations:** 1Department of Pharmacology, Pharmacognosy and Botany, Faculty of Pharmacy, Complutense University of Madrid (UCM), 28040 Madrid, Spain; 2Botany Department, EA Herbarium, National Museums of Kenya, Nairobi P.O. Box 40658-00100, Kenya; 3CNRS, Institut des Sciences Chimiques de Rennes—UMR 6226, Univ Rennes, F-35000 Rennes, France; 4Science & Education, The Field Museum, 1400 S Lake Shore Drive, Chicago, IL 60605, USA

**Keywords:** Africa, antioxidant activity, biodiversity, chemical profiling, cytotoxic activity, lichenized fungi, molecular phylogeny, species delimitation

## Abstract

We used molecular data to address species delimitation in a species complex of the parmelioid genus *Canoparmelia* and compare the pharmacological properties of the two clades identified. We used HPLC_DAD_MS chromatography to identify and quantify the secondary substances and used a concatenated data set of three ribosomal markers to infer phylogenetic relationships. Some historical herbarium specimens were also examined. We found two groups that showed distinct pharmacological properties. The phylogenetic study supported the separation of these two groups as distinct lineages, which are here accepted as distinct species: *Canoparmelia caroliniana* occurring in temperate to tropical ecosystems of a variety of worldwide localities, including America, Macaronesia, south-west Europe and potentially East Africa, whereas the Kenyan populations represent the second group, for which we propose the new species *C. kakamegaensis* Garrido-Huéscar, Divakar & Kirika. This study highlights the importance of recognizing cryptic species using molecular data, since it can result in detecting lineages with pharmacological properties previously overlooked.

## 1. Introduction

Lineages that are phylogenetically distinct but morphologically so similar that they have been included within one species are often referred to as cryptic species [1,2]. These mostly remained undiscovered before the wide application of molecular markers for phylogenetic studies that revealed that such morphology-based species may not even be closely related and in extreme cases are now classified in different genera [3,4,5]. Cryptic species are commonly found in parmelioid lichens [1,2,6], which are the largest group in the family Parmeliaceae and exhibit a remarkable diversity of secondary metabolites [7,8,9]. There has been evidence from other groups of fungi for potential qualitative and quantitative differences in the bio-activity of cryptic species [10]. Hence, we were interested in examining the pharmacological properties of the populations hypothesized to be cryptic species based on molecular data.

Lichenized fungi are known to produce several secondary metabolites that play a crucial role in systematics and taxonomy [11,12,13,14]. Lichen-derived extracts and their secondary metabolites are known to have antioxidant and cytotoxic properties [15,16,17,18,19,20,21,22]. High concentrations of reactive oxygen species (ROS) can react with cellular components such as proteins, lipids and DNA, causing a progressive structural and functional damage that results in cell death, mainly by mechanisms of apoptosis [23]. Compounds with a phenolic structure possess potential antioxidant activity by acting as hydrogen donors, singlet oxygen quenchers and reducing agents [24]. Previous studies have identified several lichen extracts and isolated compounds that scavenge or reduce ROS content, and then protect against the harmful ROS impact on health [25,26]. In addition, lichen extracts and their secondary metabolites have also shown selective cytotoxicity against different cancer cell types [27,28].

The genus *Canoparmelia* belongs to the ‘Parmotrema clade’ within the parmelioid group of the family Parmeliaceae, which represents a hyperdiverse family of lichenized fungi [8,29,30]. The species in the genus have comparatively narrow, subirregular lobes with rounded or subrounded eciliate margins, a pored epicortex, the cell walls contain isolichenan, and they have bifusiform conidia and simple rhizines [8,31]. The genus as originally circumscribed [32] was highly polyphyletic, and consequently some species were placed in other genera, such as *Austroparmelina* [3], *Crespoa* [33], and *Parmotrema* [8,34].

One of us (PKM) is working on a long-term project on the biodiversity of lichenized fungi in Kenya, where a number of *Canoparmelia* species occur. In this context, we focused on a species that morphologically resembled *C. caroliniana* but differed somewhat in its chemistry to test the hypothesis that the Kenyan samples represent a distinct clade. We used three ribosomal markers, including the nuclear internal transcribed spacer (ITS), large subunit (nuLSU), and the mitochondrial small subunit (mtSSU) to evaluate the phylogenetic relationships of the populations. In addition, we studied the chemical composition of the two *Canoparmelia* populations using HPLC_DAD_MS chromatography, and studied the pharmacological properties of the species complex.

## 2. Materials and Methods

### 2.1. Phylogenetic Analysis

Taxon sampling: The analyzed data matrices included twenty samples comprising nine species of *Canoparmelia* and two outgroup taxa. A DNA data matrix of nuLSU, ITS and mtSSU rDNA sequences was used to extrapolate evolutionary relationships. Eleven sequences were newly generated for this study. *Xanthoparmelia chlorochroa* (Tuck.) Hale and *Xanthoparmelia conspersa* (Ehrh. ex Ach.) Hale were used as outgroups. Information of studied materials, including GenBank accession numbers, is provided in Appendix A.

DNA extraction and PCR amplification: Total genomic DNA was isolated from small parts of thallus lacking any visible contamination using the USB PrepEase Genomic DNA Isolation Kit (USB, Cleveland, OH, USA). We produced sequence data from three nuclear ribosomal markers, the ITS region, a portion of the nuLSU and part of the mtSSU. Polymerase chain reaction (PCR) amplifications were executed using Ready-To-Go PCR Beads (GE Healthcare, Pittsburgh, PA, USA). Fungal ITS rDNA was amplified using primers ITS1F [35], ITS4 and ITS4A [36,37], nuLSU rDNA with LR0R and LR5 [38], and mtSSU rDNA with mrSSU1, mrSSU3R and mrSSU2R [39]. PCR products were pictured on 1% agarose gel and cleaned with ExoSAP-IT (USB, Cleveland, OH, USA). Cycle sequencing of the complementary strands was made with BigDye v3.1 (Applied Biosystems, Foster City, CA, USA) and the same primers were used for PCR amplifications. Sequenced PCR products were run on an ABI 3730 automated sequencer (Applied Biosystems) at the Pritzker Laboratory for Molecular Systematics and Evolution at the Field Museum, Chicago, IL, USA and at the Unidad de Genómica (Parque Científico de Madrid).

Sequence editing and alignment: Newly generated sequences were constructed and edited using GENEIOUS v8.1.9 [40]. Sequence alignments for each locus were executed using the program MAFFT v7 [41]. The G-INS-i alignment algorithm and ‘20PAM/K = 2′ scoring matrix, with an offset value of 0.3, were used for the ITS and nuLSU sequences. The remaining parameters were set to default values. The E-INS-i alignment algorithm and ‘20PAM/K = 2′ scoring matrix, with the remaining parameters set to default values, were used for the mtSSU sequences. The Gblocks v0.91b [42] was used to identify and eliminate ambiguous alignment nucleotide locations from the data matrix by means of the online web server (http://molevol.cmima.csic.es/castresana/Gblocks_server.html) (accessed on 20 February 2021), applying the choices for a less stringent selection of ambiguous nucleotide positions, with the ‘Allow smaller final blocks’, ‘Allow gap positions within the final blocks’, and ‘Allow less strict flanking positions’ options.

Phylogenetic analyses: The phylogenetic relationships were conducted applying maximum likelihood (ML). Investigative phylogenetic analyses of single gene tree topologies demonstrated no indication of well-supported (≥70% bootstrap values) topological conflict, therefore phylogenetic relations were assessed from a concatenated, three-locus (ITS, nuLSU, mtSSU) data matrix using a total-evidence approach [43]. We used the program RAxML v8.1.11 [44] to restructure the concatenated ML gene-tree with the CIPRES Science Gateway server (http://www.phylo.org/portal2/). We applied the HKY model, and the site heterogeneity was gamma +invariant, with locus-specific model partitions considering all loci as separate partitions. Nodal support was evaluated using 1000 bootstrap pseudoreplicates. Exploratory studies using different partitioning schemes resulted in equal topologies and comparable bootstrap support values. Phylogenetic analysis of the concatenated three-locus data matrix was also performed by applying Bayesian inference with the program BEAST v. 1.8.2. We ran two independent Markov chain Monte Carlo (MCMC) chains for 20 million generations, applying a relaxed lognormal clock and a birth–death speciation process prior. The first 25% of trees were discarded as burn-in. Chain mixing and convergence were assessed with the effective sample size (ESS) values >200 as a suitable indicator. Posterior trees from the two independent runs were combined using LogCombiner v. 1.8.0. The final maximum clade credibility (MCC) tree was assessed from the combined posterior distribution of trees.

### 2.2. Morphological and Chemical Studies

Morphological and anatomical features were studied using a Leica Wild M 8 dissecting microscope and a Leica DM RB compound microscope (Wetzlar, Germany). Chemical components were identified by high-performance thin layer chromatography (HPTLC) by means of standard methods [45], with a Camag horizontal developing chamber (Oleico Lab, Stockholm, Sweden) using solvent systems C.

In addition, 0.2 g of a panel of 13 sample thalli chosen to represent 4 species (Appendix A) was crushed under nitrogen and extracted with acetone in an ultrasonic bath. The extract was then dissolved at 1 mg/mL in acetone and filtered. HPLC analysis was carried out using a Prominence Shimadzu LC-20AD system (Marne La Vallée, France). The samples were eluted through a C18-column (2.6 µm, 100 mm × 4.6 mm, Phenomenex, Alcobendas, Spain). The mobile phase used 0.1% formic acid in HPLC-grade water and 0.1% formic acid in acetonitrile, with the gradient usually used for lichen metabolite analysis [46]. The mass spectrometry study was performed using an ADVION expression CMS mass spectrometer. We used an electrospray ionization source over a mass range of 99.85–999.8 *m*/*z* in negative mode. The ion spray voltage was set at 3.5 kV, the capillary voltage at 180 V, the source voltage offset at 20 V, the source voltage span at 20 V, the source gas temperature at 50 °C, and capillary temperature at 250 °C. MS data were obtained using Advion data express software. The datx files were transferred as *.cdf for data management using the MZmine 2.52 freeware (http://mzmine.sourceforge.net/) [47]. The mass detection was accomplished with a noise level set at 5E6. An ADAP chromatogram builder [48] was run, using as parameters: a minimal number of scans of 3, group intensity threshold and min highest intensity of 5E6, and *m*/*z* tolerance of 0.5 *m*/*z*. Deconvolution was then applied using the ADAP Wavelets algorithm. Dereplication was gathered through our laboratory database called HLDB [49], and relative quantities of the major products were determined by integration of the UV chromatograms at 254 nm.

### 2.3. Pharmacological Study

#### 2.3.1. Reagents

RPMI (Roswell Park Memorial Institute) 1640 medium, phosphate-buffered saline (PBS), dimethyl sulfoxide (DMSO), gentamicin, 3-(4,5-Dimethyl-2-thiazolyl)-2,5-diphenyltetrazolium bromide (MTT), 2,2′-azobis (2-methylpropionamidine)-dihydrochloride (AAPH), 1,1-diphenyl-2-picrylhydrazyl (DPPH), and 2,4,6-Tris (2-pyridyl)-1,3,5-triazine (TPTZ) were purchased from Sigma-Aldrich (St. Louis, MO, USA). Lichen extracts were macerated in methanol for 24 h at room temperature. Then, extracts were filtered and evaporated at room temperature.

#### 2.3.2. Antioxidant Assays

DPPH (2,2-diphenyl-1-picrylhydrazyl) radical scavenging assay: DPPH solution (50 M) and lichen extract solutions or Trolox (positive control) were mixed for 30 min in the dark at room temperature. Absorbance was then measured at 515 nm in a FLUOstar Optima fluorimeter (BMG Labtech, Ortenberg, Germany). Data are expressed as EC_50_ value [50].

Oxygen Radical Absorbance Capacity (ORAC) assay: The mixture of lichen extracts or Trolox (antioxidant reference compound) with fluorescein (70 nM) was incubated for 10 min at 37 °C. Then, a solution of 2,2′-azobis (2-amidinopropane) dihydrochloride (AAPH) (12 mM) was added. Fluorescence was measured for 98 min at λexc 485 nm and λem 520 nm using a FLUOstar Optima fluorimeter (BMG Labtech, Ortenberg, Germany). Results, calculated as the area under the fluorescein decay curve, are expressed as µmol Trolox equivalents (TE)/g DW [51].

Ferric-Reducing Antioxidant Power (FRAP) assay: This assay measures the capacity of antioxidants to reduce a ferric ion to ferrous ion. Lichen extracts were incubated with a FRAP reagent, TPTZ (2,4,6-tri(2-pyridyl)-s-triazine) (10 mM), ferric chloride (20 mM), and buffer solution (pH 3.6) for 30 min at 37 °C. Absorbance was then read at 595 nm using a microplate reader Spectrostar Nanomicroplate (BMG Labtech Inc., Ortenberg, Germany). Data are expressed as *µ*mol Fe^2+^/g extract [52].

Folin–Ciocalteu assay: This assay was used to measure the total phenolic content. Lichen extracts were incubated with Folin-Ciocalteu reagent in Na_2_CO_3_ solution (75 g/L) at dark for 1 h. Absorbance was then read at 760 nm using a microplate reader Spectrostar Nanomicroplate (BMG Labtech Inc., Ortenberg, Germany). Data are expressed as µg gallic acid equivalent (GA)/mg dry extract [53].

#### 2.3.3. Cell Culture and Cell Treatments

The human cancer cell lines MCF7 (breast cancer) and HepG2 (liver cancer) were maintained in RPMI 1640, supplemented with 10% fetal bovine serum (FBS) and gentamicin, in a humidified atmosphere at 37 °C with 5% CO_2_. Extracts were first dissolved in dimethyl sulfoxide (DMSO), and then serial dilutions in phosphate-buffered saline (PBS) were made, being the higher DMSO concentration in well/plate < 0.1%.

#### 2.3.4. Cell Viability Assay

The 3-(4,5-dimethylthiazol-2-yl)−2,5-diphenyltetrazolium bromide (MTT) colorimetric assay was employed to measure the effect of lichen extracts on cancer cells viability [54]. Cells were treated with different concentrations of lichen extracts ranged from 10 to 400 µg/mL for 24 h, and then incubated with MTT solution (2 mg/mL) for 1 h. Subsequently, DMSO was added to dissolve the formed formazan crystals. Absorbance was measured at 550 nm using a microplate reader Spectrostar Nanomicroplate (BMG Labtech Inc., Ortenberg, Germany). Data are expressed as percentage of control cells (100% of cell viability).

#### 2.3.5. Statistical Analysis

Data, representative of three independent experiments (mean ± SD), were analyzed using a one-way ANOVA followed by Tukey’s test with the statistical analysis software SigmaPlot 11.0. The significance level was *p* < 0.05.

## 3. Results and Discussion

### 3.1. Phylogenetic Analysis

A total of eleven sequences, comprising five nuclear ITS, four nuLSU and two mitochondrial SSU rDNA, were generated in this study and uploaded to GenBank (Appendix A). TNe+G, TNe+I, and HKY+I+G were selected as the best-fit models of evolution for the ITS, nuLSU, and mtSSU partitions, respectively.

No supported conflicts were found among the single-locus trees (results not shown), and therefore the concatenated three-locus data matrix was examined. The partitioned ML analysis of the concatenated data matrix resulted in an optimal tree with ln likelihood value = −4671.04 (Figure 1). Maximum likelihood and Bayesian topologies were highly similar and did not show any supported clash (e.g., PP ≥ 0.95 and ML bootstrap ≥ 70%); the ML tree topology is shown here with the Bayesian posterior probabilities included (Figure 1). We consider PP ≥ 0.95 and ML bootstrap ≥ 70% as strong support for the nodes.

The *Canoparmelia caroliniana* (Nyl.) Elix & Hale samples included in this study did not form a monophyletic group, but clustered into distantly related clades (clades 1 and 2 in Figure 1). Clade 1 forms a well-supported sister-group relationship with the isidiate species that contains usnic acid in the cortex, *C. ecaperata* (Müll. Arg.) Elix & Hale, whereas clade 2 forms a well-supported sister-group relationship with the apotheciate species *Canoparmelia austroamericana* Adler. Within clade 2, samples morphologically agreeing with *C. amabilis* Heiman & Elix and *C. caroliniana* clustered together, supporting previous results of conspecificity of these two taxa [55]. This further supports previous studies [34,56] that suggest the species delimitation in *Canoparmelia* needs attention. As the sample of *C. caroliniana* from the type locality, i.e., North Carolina (Appendix A), belongs to clade 2, we here consider this clade to be *C. caroliniana* s. str. Consequently, the Kenyan samples grouped in clade 1 is accommodated in a new species, *C. kakamegaensis*.

### 3.2. Chemical and Morphological Analysis

We re-examined the secondary chemistry and morphology of the samples of both clades of *C. caroliniana*. Both clades contain minor amounts of atranorin as a cortical component, and as medullary substances, with stenosporic and perlatolic acids as major, and glomelliferic acid as submajor substances. The quantity of the substances differed somewhat between the two clades, with an average percentage of stenosporic acid (46.4%, SD ± 3.34) being lower in clade 2 than clade 1 (60.2%), whereas perlatolic acid was present in higher percentages in clade 2 (32.6%, SD ± 5.11) than in clade 1 (23.6%). However, our sampling is insufficient and additional studies are required to test whether chemosyndromic variation [57] occurs in this species complex. While *C. amabilis* has a similar chemical profile to *C. caroliniana*, a clear distinction with *Canoparmelia concrescens* (Vain.) Elix & Hale can be based on, in the former, the presence of divaricatic acid as a major compound, with stenosporic, subdivaricatic, and nordivaricatic as minor compounds, in addition to a lack of glomelliferic acid and atranorin (Table 1). Of note, looking at the chemical profile of historical herbarium samples (collected over 60 years ago), it is noticeable to observe a similar profile to freshly collected materials (Appendix A). This indicates a long-time stability of these compounds, and furthers incentives to recognize their heritage value, as new techniques are suitable for analyses. However, additional study will be needed to challenge this hypothesis.

A re-examination of morphological features of samples from both clades revealed the width of lobes as the only distinguishing morphological characteristic. Based on the subtle morphological differences and the results of the phylogenetic analysis inferred from ribosomal markers, we propose a new species to accommodate specimens of clade 1 below.

### 3.3. Taxonomy

***Canoparmelia kakamegaensis*** Garrido-Huéscar, Divakar & Kirika, **sp. nov**. (Figure 2).

MycoBank No. MB844996.

Diagnosis: Morphologically similar to *C. caroliniana* but differs in having narrower lobes.

Type: KENYA. Kakamega County., Kakamega forest, Isecheno Forest Station, 1760 m, 0°14′ N 034°52′ S, on bark, 05 August 2013, P. Kirika 3419 (holotype: EA, isotypes: F, MAF) (Figure 2).

Genbank accession numbers: ITS and nuLSU; KX369243 and KX369261.

Etymology: The taxon name is based on its occurrence in the Kakamega forest in Kenya.

Description: Thallus foliose, adnate, pale grey. Lobes 1-3 mm broad, irregularly incised, subirregularly branched, crenate, eciliate. Upper cortex more or less maculate, cracked, isidiate. Medulla white. Underside black with a narrow, brown marginal zone, rhizines simple, black. Isidia laminal, slender, cylindrical, mostly simple or rarely branched, concolorous with the upper surface. Apothecia and pycnidia absent.

Secondary chemistry: Atranorin in cortex, in medulla stenosporic and perlatolic acids as major, and glomelliferic acid as submajor substances. Furthermore, along with minor amounts of atranorin, traces of compounds are detected and putatively identified as olivetolcarboxylic, 4-O-methylolivetoric, and divaricatic acids.

Ecology: Corticolous in tropical rain forest, 1760 m altitude.

Notes: *Canoparmelia kakamegaensis* can easily be confused with *C. caroliniana* in the field, but the former has narrower lobes. In spite of morphological similarities, *C. kakamegaensis* does not form a sister-group relationship with *C. caroliniana* but is a sister group to *C. ecaperata* (Figure 1). Conversely to *C. ecaperata*, usnic acid is lacking in *C. kakamegaensis*.

Additional specimen examined: KENYA. Kakamega County., Kakamega forest, Buyangu, Edeos Campsite, tropical rain forest, 1758 m, 00°21′ N, 034°51′ E, 31 July 2013, P. Kirika 3389 (EA).

### 3.4. Pharmacological Study

The antioxidant activities of lichen methanol extracts were investigated using in vitro assays including DPPH, ORAC and FRAP methods (Table 2).

The antioxidant potency of the *C. kakamegaensis* extract was more effective in DPPH and ORAC antioxidant assays than *C. caroliniana*. Statistical differences were found between the antioxidant activity (*p* < 0.05). Particularly, DPPH revealed that the EC50 value for the *C. kakamegaensis* extract was 400.2 µg/mL, whereas the value for *C. caroliniana* was 1004.3 µg/mL. Moreover, the *C. kakamegaensis* extract showed a higher ORAC value than *C caroliniana* (3.2 µmol TE/mg dry extract and 1.3 µmol TE/mg dry extract, respectively). Regarding the ability to reduce the TPTZ–Fe (III) complex to TPTZ–Fe (II), the FRAP value was similar for both species. Finally, although there are slight differences between *C. kakamegaensis* and *C. caroliniana* in the content of the secondary metabolites stenosporic and perlatolic acids (with slightly higher content in the former), there is a higher value in the total phenolic content for *C. kakamegaensis* (53.7 µg GA/mg of the dry extract versus 44.7 µg GA/mg of the dry extract).

We assessed the effect of the two lichen methanol extracts on the cell viability of the human breast MCF7 cancer cell line and on the human liver HepG2 cancer cell line. For this purpose, these cancer cells were treated with a range of lichen extracts concentrations from 10 to 400 µg/mL for 24 h. As shown in Figure 3, both lichens inhibited cancer cell growth in a concentration-dependent manner.

We then calculated IC50 value (concentration that causes 50% of cell growth inhibition). Although slight, a significant difference was observed between *C. kakamegaensis* and *C. caroliniana* activities for better activity on both cancer cell lines (Appendix A).

In the current work, we evaluated the in vitro antioxidant and cytotoxic activities of the methanol extracts of *Canoparmelia kakamegaensis* and *C. caroliniana.* Lichens have received great interest in recent years because they contain unique compounds with potential, especially antioxidant and cytotoxic activity [58]. The biological activity of the well-known lichen *C. caroliniana* has not been previously reported.

Compounds with antioxidant capacity are of therapeutic interest due to their ability to prevent biological damage from the overproduction of free radicals [59]. The antioxidant activity was investigated using different in vitro assays that differ in their mechanism of action. The DPPH and FRAP assays are single-electron transfer mechanism (SET)-based. These methods measure how antioxidants are able to transfer an electron to a free radical. In contrast, the ORAC assay is based on hydrogen atom abstraction mechanisms (HAT), which involve the direct abstraction of a hydrogen atom from an antioxidant to a free radical [60]. In the current work, our findings showed that the methanol extract of *Canoparmelia kakamegaensis* had a higher in vitro antioxidant effect than *C. caroliniana* in DPPH and ORAC assays. Moreover, a positive correlation between the total phenolic content and the in vitro antioxidant activity was observed, which is very similar to what was reported previously [21].

The cytotoxic activity of the methanol lichen extracts was examined using MCF7 and HepG2 cancer cell lines. Breast cancer cells are the most common malignancy in females (23% of diagnosed total cancer cases and 14% of total deaths attributed to cancer). The breast MCF7 cell line is widely used in research for estrogen receptor (ER)-positive breast cancer [61]. On the other hand, liver cancer constitutes the fifth most common type of cancer, and the third most common cause of death due to cancer. In our study, *Canoparmelia kakamegaensis* had higher cytotoxic activity against both cancer cell lines compared to the activity of *C. caroliniana.* According to the American National Cancer Institute guidelines, both methanol lichen extracts showed moderate cytotoxic activity (IC50 > 30 µg/mL) against both cancer cell lines [62]. Whether the differences found in the pharmacological properties of *C. caroliniana* and *C. kakamegaensis* are due to the different quantities of stenosporic and perlatolic acids, or other differences, remains to be seen.

## Figures and Tables

**Figure 1 jof-08-00826-f001:**
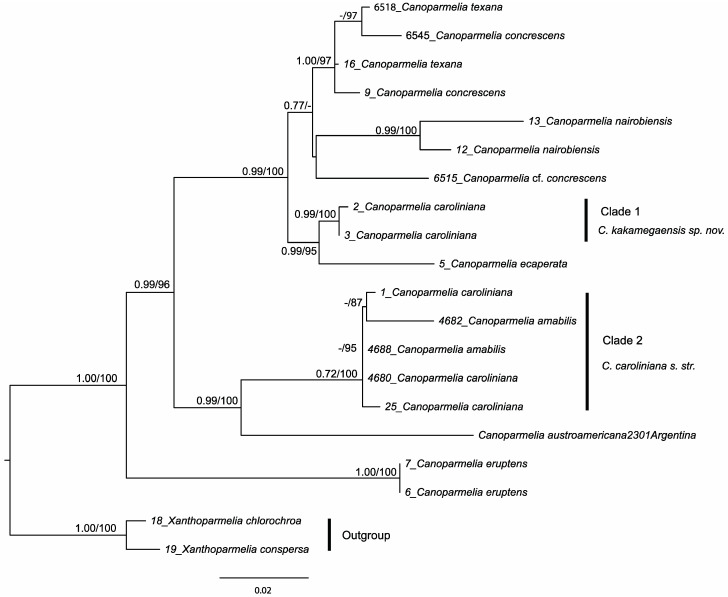
The phylogenetic relations of *Canoparmelia* species based on maximum likelihood (ML) and Bayesian analyses of a concatenated, three-locus data set (ITS, nuLSU and mtSSU rDNA). The ML tree is depicted here. Posterior probabilities ≥ 0.95 from the Bayesian analysis and ML bootstrap values ≥ 70% are provided above branches.

**Figure 2 jof-08-00826-f002:**
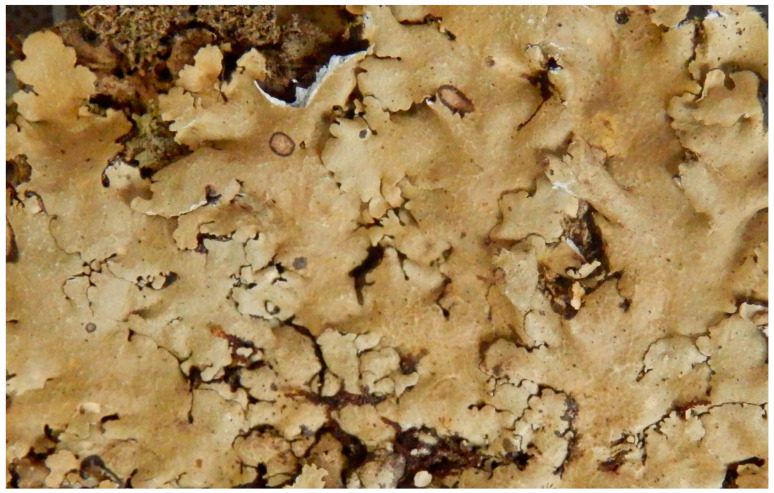
*Canoparmelia kakamegaensis*, habit (P. Kirika 3419).

**Figure 3 jof-08-00826-f003:**
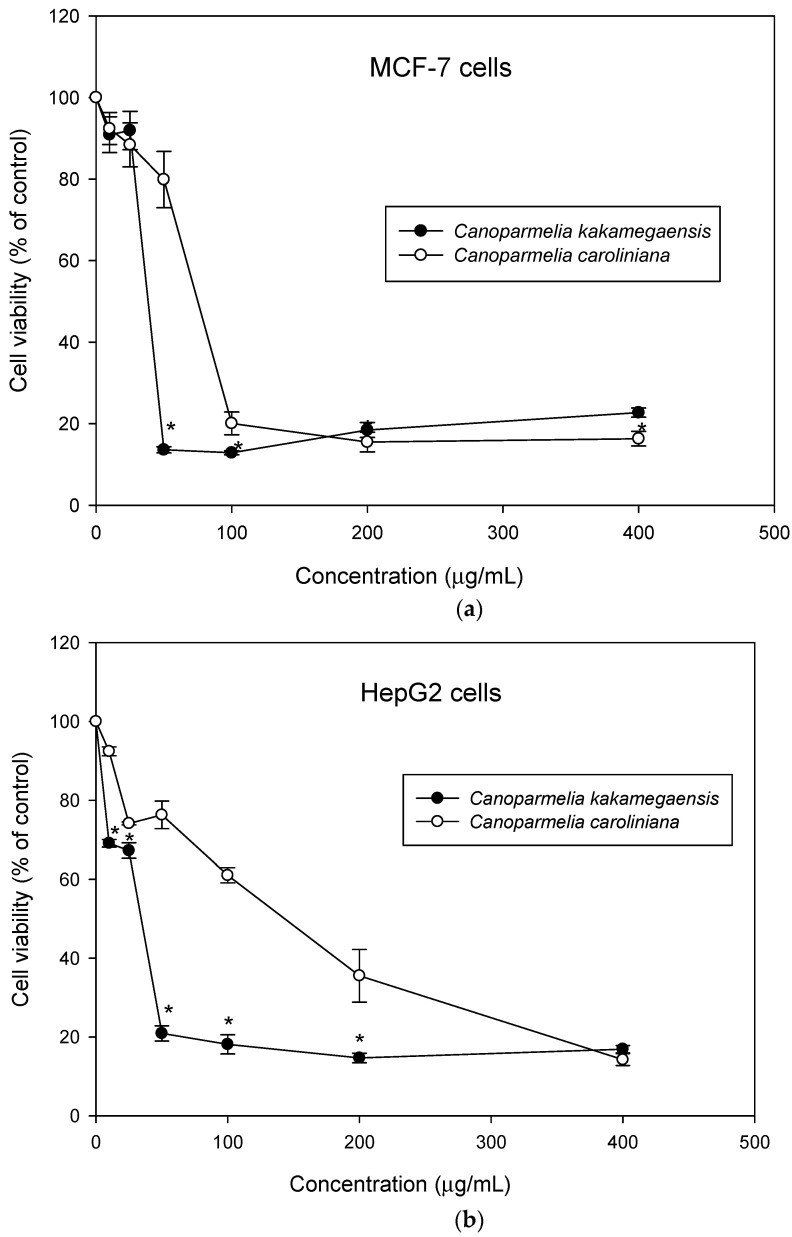
Effect on cell viability (**a**) MCF-7 cells and (**b**) HepG2 cells treated with different concentrations of extracts of *Canoparmelia kakamegaensis* and *Canoparmelia caroliniana* for 24 h using MTT assay. Values are expressed as mean ± standard deviation. * Indicates statistically significant differences (*p* < 0.05) between lichen extracts.

**Table 1 jof-08-00826-t001:** Relative amount percentages of metabolites found in *Canoparmelia caroliniana*, *C. amabilis*, *C. kakamegaensis* and C. *concrescens*, based on HPLC-DAD analysis.

	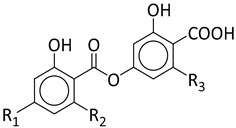						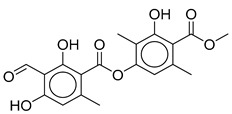
	Stenosporic Acid	Perlatolic Acid	Glomelliferic Acid	Divaricatic Acid	Nordivaricatic Acid	Subdivaricatic ACID	Atranorin
R_1_	OCH_3_	OCH_3_	OCH_3_	OCH_3_	OH	OCH_3_	
R_2_	C_3_H_7_	C_4_H_9_	(CH_2_-CO-C_3_H_7_)	C_3_H_7_	C_3_H_7_	CH_3_	
R_3_	C_4_H_9_	C_4_H_9_	C_4_H_9_	C_3_H_7_	C_3_H_7_	C_3_H_7_	
*C. amabilis* (*n* = 2)	45.8	34.5	16.8	-	-	-	1.9
*C. caroliniana* (*n* = 8)	46.2	31.8	15.9	-	-	-	5.6
*C. kakamegaensis **	60.2	23.6	15.8	-	-	-	0.4
*C. concrescens* (*n* = 2)	4.9	-	-	85.3	4.5	5.3	-

* Due to shortage of materials, analysis was only performed on one sample.

**Table 2 jof-08-00826-t002:** Extraction yield, total phenolic content and antioxidant potential (DPPH, ORAC and FRAP assays) of the lichen extracts *Canoparmelia kakamegaensis* and *Canoparmelia caroliniana*. Values are expressed as mean ± standard deviation. * Indicates statistically significant differences (*p* < 0.05) between lichen extracts.

Lichen Species	Yields (% *w/w*)	Total Phenolic Contents (µg GA/mg)	DPPH EC50 (µg/mL)	ORAC Value (µmol TE/mg Dry Extract)	FRAP (µmol of Fe^2+^ eq/g Sample)
*Canoparmelia kakamegaensis*	18.5 ± 2.0	53.7 ± 2.8 *	400.2 ± 34.9 *	3.2 ± 0.1 *	9.6 ± 0.2
*Canoparmelia caroliniana*	4.9 ± 0.7	44.7 ± 2.2	1004.3 ± 112.8	1.3 ± 0.04	10.8 ± 1.0

## Data Availability

Newly generated sequences have been deposited to a public repository i.e., GenBank. Further, all the data used to develop this manuscript are available through Appendix A.

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
