# Peer review of "A New Cryptic Lineage in Parmeliaceae (Ascomycota) with Pharmacological Properties"

_jof, 2022, doi:10.3390/jof8080826_

Round 1

Reviewer 1 Report

The manuscript uses a polyphasic approach (Morphological characters, molecular phylogeny and pharmacological properties) to delimite the new species Canoparmelia kakamegaensis. In general, all sections of manuscript are clearly explained. The subject if of interest for mycologists. Thus, it is suitable to be published in the Journal of Fungi. Below are some minor errors for the authors' consideration.

General comment:

The manuscript uses a polyphasic approach (Morphological characters, molecular phylogeny and pharmacological properties) to delimite the new species Canoparmelia kakamegaensis. In general, all sections of manuscript are clearly explained. The subject if of interest for mycologists. Thus, it is suitable to be published in the Journal of Fungi. Below are some minor errors for the authors' consideration.

 Minor format corrections:

1. Page 1, line 36 (Introduction). Please modify the sentence “… the family Parmeliaceae and exhibits…” by “…the family Parmeliaceae and exhibit…”

2. Page 2, line 54 (Introduction). Please modify the sentence “… which represents a hyperdiverse diverse family…” by “…which represents a hyperdiverse family…”

3. Page 2, line 74 (Materials and Methods). Please modify the sentence “18 sequences were…” by “Eighteen sequences were…”

4. Page 2, line 75 (Materials and Methods). Please provide the complete name of outgroup taxa, Xanthoparmelia chlorochora and X. conspersa including the namers, when they appear the first time in the Materials and Methods text. Further mentions within the same abstract can be abbreviated.

5. Page 2, line 84 (Materials and Methods). Please delete the sentence “…using dilutions of total DNA.”

6. Page 2, line 84 (Materials and Methods). Please modify the sentence "Fungal ITS rDNA …the primers mrSSU1, mrSSU3R and mrSSU2R" .

7. Page 2, line 89 (Materials and Methods). Please modify the sentence "… performed using BigDye v3.1 …  and the same primers used for …" by "… performed using BigDye v3.1 …  and the same primers were used for …".

8. Page 3, line 120 (2.2. Morphological and chemical studies). Please add the country and city of microscope.

9. Page 3, line 133 (2.2. Morphological and chemical studies). Please modify the sentence "… was set at 3.5kV," by "… was set at 3.5 kV,"

10. Page 4, line 174 (2.3.2. Antioxidant assays). Please modify the sentence "… Folin-Ciocalteu assay is used… " by " … Folin-Ciocalteu assay was used… "

11. Page 4, line 200 (2.3.5. Statistical analysis). Please modify the sentence "…p<0.05… " by " …p < 0.05… "

12. Page 5, line 214 (Phylogenetic analysis). Please modify the sentence " We consider PP ≥0.95…" by " We consider PP ≥ 0.95… "

13. Page 5, line 219 (Phylogenetic analysis). Please provide the complete name of C. austroamericana.

14. Page 6, line 235 (3.2. Chemical and morphological analysis). Please modify the sentence "… Canoparmelia caroliniana." by "… C. caroliniana."

15. Page 6, line 242 (3.2. Chemical and morphological analysis). Canoparmelia amabilis can be abbreviated.

16. Page 6, line 243 (3.2. Chemical and morphological analysis). Please provide the complete name of C. concrescens.

17. Page 7, line 266 (3.3 Taxonomy). Please modify "1760m" by "1760 m"

18. Page 8, line 271 (Description). Please modify "Lobes 1-3mm broad" by " Lobes 13 mm broad".

19. Page 8, line 271 (Ecology). Please modify "1760m" by " 1760 m ".

20. Page 8, line 287 (Additional specimen examined). Please modify "1758m" by " 1758 m ".

21. Page 9, line 296, 336 and 347 (3.3 Pharmacological study). Please modify "…Canoparmelia kakamegaensis…" by " …C. kakamegaensis… "

22. Page 11, line 348  (3.3 Pharmacological study). Please modify the sentence "…had a higher in vitro antioxidant effect than to C. caroliniana…" by " …had a higher in vitro antioxidant effect than C. caroliniana… "

23. Page 15, line 507 (Reference 52). Please modify the reference and add the pages.

24. Page 15, line 512 (Reference 55). Please modify the name of Canoparmelia carolinana by italic type.

25. Page 15, line 515 (Reference 56). Please add the pages

26. Page 15, line 519 (Reference 58). Please confirm the reference.

Author Response

REVIEWER 1:

General comment:

The manuscript uses a polyphasic approach (Morphological characters, molecular phylogeny and pharmacological properties) to delimite the new species Canoparmelia kakamegaensis. In general, all sections of manuscript are clearly explained. The subject if of interest for mycologists. Thus, it is suitable to be published in the Journal of Fungi. Below are some minor errors for the authors' consideration.

Response: We highly appreciate for your valuable comments and suggestions and that have been very helpful to us to improving the manuscript.

Minor format corrections:

  1. Page 1, line 36 (Introduction). Please modify the sentence “… the family Parmeliaceae and exhibits…” by “…the family Parmeliaceae and exhibit…”

Response: The sentence “… the family Parmeliaceae and exhibits…” has been changed by “…the family Parmeliaceae and exhibit…” in the revised manuscript.

  1. Page 2, line 54 (Introduction). Please modify the sentence “… which represents a hyperdiverse diverse family…” by “…which represents a hyperdiverse family…”

Response: The sentence “… which represents a hyperdiverse diverse family…” has been changed by “…which represents a hyperdiverse family…” in the revised manuscript.

  1. Page 2, line 74 (Materials and Methods). Please modify the sentence “18 sequences were…” by “Eighteen sequences were…”

Response: The sentence “18 sequences were…” has been changed by “Eighteen sequences were…” in the revised manuscript.

  1. Page 2, line 75 (Materials and Methods). Please provide the complete name of outgroup taxa, Xanthoparmelia chlorochora and X. conspersa including the namers, when they appear the first time in the Materials and Methods text. Further mentions within the same abstract can be abbreviated.

Response: As suggested, the complete name with authors of the two Xanthoparmelia species have been provided.

  1. Page 2, line 84 (Materials and Methods). Please delete the sentence “…using dilutions of total DNA.”

Response: The sentence “… using dilutions of total DNA” has been deleted in the revised manuscript.

  1. Page 2, line 84 (Materials and Methods). Please modify the sentence "Fungal ITS rDNA …the primers mrSSU1, mrSSU3R and mrSSU2R" .

Response: The sentence has been modified.

Modified text: “Fungal ITS rDNA was amplified using primers ITS1F [35], ITS4 and ITS4A [36,37]; nuLSU rDNA with LR0R and LR5 [38], and mtSSU rDNA with mrSSU1, mrSSU3R and mrSSU2R”.

  1. Page 2, line 89 (Materials and Methods). Please modify the sentence "… performed using BigDye v3.1 …  and the same primers used for …" by "… performed using BigDye v3.1 …  and the same primers were used for …".

Response: The sentence "… performed using BigDye v3.1 …  and the same primers used for …" has been changed by "… performed using BigDye v3.1 …  and the same primers were used for …" in the revised manuscript.

  1. Page 3, line 120 (2.2. Morphological and chemical studies). Please add the country and city of microscope.

Response: The city and country are added.

  1. Page 3, line 133 (2.2. Morphological and chemical studies). Please modify the sentence "… was set at 3.5kV," by "… was set at 3.5 kV,"

Response: The sentence "… was set at 3.5kV," has been changed by "… was set at 3.5 kV," in the revised manuscript.

  1. Page 4, line 174 (2.3.2. Antioxidant assays). Please modify the sentence "… Folin-Ciocalteu assay is used… " by " … Folin-Ciocalteu assay was used… "

Response: The sentence "… Folin-Ciocalteu assay is used… " has been changed by " … Folin-Ciocalteu assay was used… " in the revised manuscript.

  1. Page 4, line 200 (2.3.5. Statistical analysis). Please modify the sentence "…p<0.05… " by " …p < 0.05… "

Response: The sentence "…p<0.05… " has been changed by " …p < 0.05… "  in the revised manuscript.

  1. Page 5, line 214 (Phylogenetic analysis). Please modify the sentence " We consider PP ≥0.95…" by " We consider PP ≥ 0.95… "

Response: The sentence " We consider PP ≥0.95…" has been changed by " We consider PP ≥ 0.95… " in the revised manuscript.

  1. Page 5, line 219 (Phylogenetic analysis). Please provide the complete name of C. austroamericana.

Response: The full name of Canoparmelia austroamericana with authority is given.

  1. Page 6, line 235 (3.2. Chemical and morphological analysis). Please modify the sentence "… Canoparmelia caroliniana." by "… C. caroliniana."

Response: The sentence "… Canoparmelia caroliniana." has been changed by "… C. caroliniana." in the revised manuscript.

  1. Page 6, line 242 (3.2. Chemical and morphological analysis). Canoparmelia amabilis can be abbreviated.

Response: "Canoparmelia amabilis" has been abbreviated in the revised manuscript.

  1. Page 6, line 243 (3.2. Chemical and morphological analysis). Please provide the complete name of C. concrescens.

Response: The full name of Canoparmelia concrescens with authority is given.

  1. Page 7, line 266 (3.3 Taxonomy). Please modify "1760m" by "1760 m"

Response: The sentence "1760m" has been changed by "1760 m" in the revised manuscript.

  1. Page 8, line 271 (Description). Please modify "Lobes 1-3mm broad" by " Lobes 1–3 mm broad".

Response: The sentence "Lobes 1-3mm broad" has been changed by " Lobes 1–3 mm broad" in the revised manuscript.

  1. Page 8, line 271 (Ecology). Please modify "1760m" by " 1760 m ".

Response: The sentence "1760m" has been changed by " 1760 m " in the revised manuscript.

  1. Page 8, line 287 (Additional specimen examined). Please modify "1758m" by " 1758 m ".

Response: The sentence "1758m" has been changed by " 1758 m " in the revised manuscript.

  1. Page 9, line 296, 336 and 347 (3.3 Pharmacological study). Please modify "…Canoparmelia kakamegaensis…" by " …C. kakamegaensis… "

Response: The sentence "…Canoparmelia kakamegaensis…"  has been changed by " …C. kakamegaensis… " in the revised manuscript.

  1. Page 11, line 348  (3.3 Pharmacological study). Please modify the sentence "…had a higher in vitro antioxidant effect than to C. caroliniana…" by " …had a higher in vitro antioxidant effect than C. caroliniana… "

Response: The sentence "…had a higher in vitro antioxidant effect than to C. caroliniana…"  has been changed by " …had a higher in vitro antioxidant effect than C. caroliniana… " in the revised manuscript.

  1. Page 15, line 507 (Reference 52). Please modify the reference and add the pages.

Response: The reference has been modified and add the pages in the revised manuscript.

  1. Page 15, line 512 (Reference 55). Please modify the name of Canoparmelia carolinana by italic type.

Response: Canoparmelia caroliniana has been written in italics in the revised manuscript.

  1. Page 15, line 515 (Reference 56). Please add the pages

Response: The paper is in press and thus the doi number is given.

  1. Page 15, line 519 (Reference 58). Please confirm the reference.

Response: We confirm the reference.

Reviewer 2 Report

The study is well-prepared and described, the outcome is interesting and useful for the scientific community. I consider, the paper can be accepted in its present form. However, I suggest the editorial correction before publishing the paper. 

Lines 72-72: "Taxon sampling - The analyzed data matrices included 20 samples comprising nine species of Canoparmelia and two outgroup taxa. (...) 18 sequences were newly generated for this study. "

Please, add the locations of sampling areas for 18 new sequences.

In addition, the sentence cannot begin with a number. The number 18 should be replaced by the word "eighteen". (Please, look at: https://www.grammar-monster.com/lessons/numbers_starting_sentences.htm )

Line 264: "MycoBank No. xxxx"

Please, add the MycoBank number. Moreover, please check and complete all missing links.

Author Response

REVIEWER 2:

The study is well-prepared and described, the outcome is interesting and useful for the scientific community. I consider, the paper can be accepted in its present form. However, I suggest the editorial correction before publishing the paper. 

Response: We thank you very much for your feedback and wonderful suggestions that have been very helpful for us.

Lines 72-72: "Taxon sampling - The analyzed data matrices included 20 samples comprising nine species of Canoparmelia and two outgroup taxa. (...) 18 sequences were newly generated for this study. "

Please, add the locations of sampling areas for 18 new sequences.

Response: The location of the sampling area and GenBank accession numbers of the newly generated sequences were added.

In addition, the sentence cannot begin with a number. The number 18 should be replaced by the word "eighteen". (Please, look at: https://www.grammar-monster.com/lessons/numbers_starting_sentences.htm )

Response: Done as suggested.

Line 264: "MycoBank No. xxxx"

Please, add the MycoBank number. Moreover, please check and complete all missing links.

Response: MycoBank number “MB844996” is added. Further, we revised the missing links and included GenBank accession numbers of newly generated sequences.